# Intranasal Immunization with DNA Vaccine HA-CCL19/Polyethylenimine/Chitosan Composite Provides Immune Protection Against H7N9 Infection

**DOI:** 10.3390/vaccines13010010

**Published:** 2024-12-26

**Authors:** Yuqing Xiang, Hongbo Zhang, Youcai An, Ze Chen

**Affiliations:** 1Department of Basic Research, Ab & B Bio-Tech Co., Ltd. JS, Taizhou 225300, China; xiangyuqing@abbbio.com.cn (Y.X.); anyoucai@abbbio.com.cn (Y.A.); 2College of Life Science, Hunan Normal University, Changsha 410081, China; 3Innovative Human Vaccine Technology and Engineering Research Center of Taizhou, Taizhou 225300, China; 4Innovative Antiviral Vaccines Engineering Technology Research Center of Taizhou, Taizhou 225300, China

**Keywords:** influenza, HA, CCL19, polyethyleneimine, chitosan, HA-CCL19/poly-ethylenimine/chitosan

## Abstract

Background/Objectives: The H7N9 avian influenza virus (AIV) constitutes a novel subtype of influenza virus that has emerged within the past decade. Empirical studies have demonstrated that H7N9 AIV holds the potential to trigger a human pandemic. Vaccines constitute the sole armament available to humanity in combating influenza epidemics. DNA vaccines present numerous merits; however, substantial conundrums persist regarding how to augment their immunogenicity and implement their delivery through mucosal immunization. Methods: In this study; BALB/c mice were utilized as a model to investigate the effect of CCL19 as a molecular adjuvant and to determine the immune response elicited by polyethylene imine (PEI) and chitosan (CS) as adjuvants during the delivery of a DNA vaccine through the nasal mucosal route. Results: Our results revealed that the CCL19 molecular adjuvant exerts a substantial immunomodulatory enhancement effect on the H7N9-HA DNA vaccine, inducing more pronounced cellular and humoral immunity. Additionally, our results indicated that the composite formed by the HA-CCL19 DNA in combination with PEI and CS effectively activates local mucosal immunity as well as systemic humoral and cellular immunity, offering 100% protection against lethal doses of homologous virus challenges. Conclusions: CCL19 conspicuously augments the immunogenicity of the influenza virus HA DNA and conserves the integrity of the vaccine antigen. Simultaneously, CS and PEI proficiently facilitate the mucosal delivery of DNA, thereby eliciting mucosal immunity related to DNA vaccines. This study investigated the feasibility of utilizing nasal mucosa for DNA vaccine immunization, which holds significant implications for the advancement and application of DNA vaccines in public health

## 1. Introduction

The avian influenza virus (AIV) demonstrates pronounced mutability, engendering the advent of novel subtypes of influenza viruses with the potential to traverse species boundaries and infect humans [1]. In recent years, human cases of infection with the H7N9 AIV have been reported. The low-pathogenic H7N9 AIV, initially isolated from humans in 2013, has evolved to acquire the key attributes of highly pathogenic AIV by 2017. This virus poses a severe threat to the life and health of both animals and humans [2]. Some researchers predict that this subtype of AIV might trigger a pandemic among humans.

Vaccination constitutes an efficacious modality for precluding influenza and its attendant complications, especially for individuals at heightened risk [3]. Owing to its facile preparation, DNA vaccines offer convenience in vaccine research [4,5]. The investigation of molecular adjuvants and vaccination routes has commanded the focus of immunologists [6,7,8,9,10]. Cytokines have been ascertained to exhibit robust adjuvant effects in vaccines [11,12,13,14]. They can be employed to concurrently express cytokines along with protein vaccines, to generate recombinant proteins, or to construct recombinant plasmids of cytokines for probing their adjuvant efficacy in DNA vaccines [15]. Chemokine (C-C motif) ligand 19 (CCL19) has been verified to elicit the rapid endocytic uptake of antigens by dendritic cells (DCs) and augment the survival of mature DCs. Moreover, it has been reported that CCL19 induces dendritic elongation in DCs, thereby facilitating interaction with T cells [16,17,18,19]. Additionally, the exploration of immune pathways, particularly mucosal immunity, holds vast prospects in clinical transformation, and mucosal immunity has drawn escalating attention from researchers [15,20,21,22].

Chitosan (CS) and polyethyleneimine (PEI) exhibit bio-adhesive and immunomodulatory properties, making them widely utilized as adjuvants and delivery vectors in the field of mucosal immunity research [23,24]. Both PEI and CS can bind to proteoglycans on the cell surface, facilitating their entry into cells via endocytosis. Previous studies have demonstrated that PEI-modified chitosan exhibits high transfection efficiency in vivo [25]. Intranasal immunization presents several advantages over intramuscular immunization, including enhanced immunization efficiency and improved patient compliance. The investigation of integrating the benefits provided by intranasal immunization with DNA vaccines holds significant implications for public health.

In this investigation, a murine model was utilized to explore the function of CCL19 as an adjuvant in potentiating the immunogenicity of the H7N9 AIV hemagglutinin (HA) DNA vaccine. Moreover, polyethyleneimine (PEI) and a chitosan/HA-CCL19 DNA vaccine composite was meticulously evaluated to assess their protective efficacy against influenza virus challenges via nasal mucosal immunization.

## 2. Materials and Methods

### 2.1. Viruses and Plasmids

Virus: The influenza virus strain employed in this study was mouse-adapted NIBRG-267 (H7N9).

Plasmid construction: The RNA of the H7N9 virus was extracted using the TRIzol (Invitrogen, Waltham, MA, USA, Catalog Numbers: 15596026), and complementary DNA (cDNA) was synthesized through reverse transcription employing commercial kits (Takara, Dalian, China, Cat# 6110A). The primer sequences used for PCR amplification of the hemagglutinin (HA) gene were as follows: HA-F (with *XhoⅠ* site): GGCTCGAGATGACTCAAATCCTGGTATT; HA-R (with *NotⅠ* site): TCGCGGCCGCTTATATACAAATAGTGCACCGCAT. They were synthesized by Genewiz (Suzhou, China). Subsequently, both the HA PCR product and pCAGGS vectors were digested with *XhoⅠ* and *NotⅠ* restriction enzymes, followed by ligation using T4 DNA ligase.

The coding sequence of the HA gene of the H7N9 virus was concatenated to the murine CCL19 gene sequence by means of a flexible GGGGS linker, and the entire sequence was synthesized by Genewiz (Suzhou, China) and subsequently cloned into the pCAGGS vector.

### 2.2. Animal Immunization and Challenge

Specific pathogen-free (SPF) female BALB/c mice, aged 6 to 8 weeks, were procured from the Center for Disease Control and Prevention in Hubei Province, China, and were maintained under SPF conditions. All experiments involving animals have been approved by the Animal Care Committee of Ab & B Bio-Tech Co., Ltd. JS, Taizhou (ethical approval number: Ab&B-IV2022008; date of approval: 6 January 2022). The ARRIVE guidelines were followed for all the mouse experiments in this study.

The mice were subjected to immunization through intramuscular injection in conjunction with electroporation, employing a voltage of 100 V for a duration of 50 ms for three consecutive stimulations, followed by an additional three repetitions after polarity reversal with a one-second interlude. Regarding the intranasal immunization study, the mice were anesthetized first, and a 20 µL volume consisting of the DNA vaccine and a complex of polyethyleneimine (PEI) and chitosan (CS) was instilled into their nostrils.

Two weeks subsequent to primary immunization, blood samples were collected for serum separation to determine IgG along with IgG1 and IgG2a levels. Two weeks following booster immunization, splenic lymphocytes were harvested for cytokine profiling. The mice were then challenged with a lethal H7N9 virus at a dose equivalent to 5 LD_50_. On the third day subsequent to the challenge, lung tissue homogenates were obtained for viral titer determination. The remaining mice were monitored daily, and their morbidity and mortality rates were meticulously recorded. When the mice underwent a body weight reduction surpassing 25%, they were subjected to euthanasia (Figure 1).

### 2.3. Fabrication of PEI-CS Composite

PEI (branched, average Mw~25000, Sigma-Aldrich, St. Louis, MO, USA, Cat# 408727) was dissolved in ultrapure water to achieve a final concentration of 1 mg/mL. An aqueous solution of 1% acetic acid was formulated first, and subsequently, a 0.1% chitosan (deacetylated ≥ 75%, Sigma-Aldrich, St. Louis, MO, USA, Cat# C3646-10G) solution (mass-to-volume ratio) was prepared with the 1% aqueous acetic acid solution. First, the required PEI aqueous solution was prepared based on the total amount of DNA needed for immunizing the mice, adhering to a PEI/DNA N/P ratio of 10. N/P refers to the ratio of moles of amine groups in cationic polymers to those of phosphate groups in DNA. The N/P ratio is calculated by the formula 7.53 × b/c, where b represents the mass of the PEI (μg) and c denotes the mass of the plasmid (μg). This mixture was then added to the DNA aqueous solution, where N signifies the amino groups of PEI and P indicates the phosphate groups of DNA. The components were thoroughly mixed with DNA and allowed to stand at room temperature for 30 min to facilitate the synthesis of the DNA-PEI nanocomposite. Subsequently, this complex was preheated to 55 °C in conjunction with a 0.1% CS solution. The corresponding composite were obtained by rapidly adding an equal volume of CS solution to the DNA-PEI complex. The mixture was vortexed for 30 s and left at room temperature for an additional 30 min.

### 2.4. Quantitative Analysis of Antibodies Through ELISA

The serum specimens were segregated from the blood and employed for IgG, IgG1, and IgG2a assays, respectively. Nasal lavages were utilized for secretory immunoglobulin A (sIgA) determination. The quantities of antibodies against HA were quantified by enzyme-linked immunosorbent assay (ELISA) as previously delineated [26].

### 2.5. Hemagglutination Inhibition Assay

The hemagglutination inhibition assay was performed as described previously [27]. Briefly, serum separated from mice was treated with receptor-destroying enzyme (RDE) (Denka Seiken, Tokyo, Japan, 340016) first. The treated serum was thoroughly amalgamated with a pre-prepared working solution comprising 4 hemagglutination units (HAU) and was permitted to incubate at room temperature for 30 min, as formerly delineated in our publication. Subsequently, a 0.5% suspension of chicken red blood cells was added, followed by gentle agitation and an additional incubation period of 30 min at room temperature. The titers of the hemagglutination inhibition titers were determined and recorded determined as the highest serum dilution that completely inhibited hemagglutination.

### 2.6. Cytokine Assays

Splenocytes harvested from immunized mice were isolated and cultivated. Subsequent to stimulation with the lysate of purified H7N9 virus, the supernatants of the cell cultures were collected, and the concentrations of cytokines INF-γ and IL-2 were determined using the ELISA kit (Dakewe, Shenzhen, China, Cat# 1210002 & 120203).

### 2.7. Statistical Analysis

In this study, antibody titers and weight loss rates were subjected to one-way analysis of variance (ANOVA) in GraphPad Prism, when there is only one independent variable with more than two levels or groups. The survival rates of different groups were appraised by the log-rank test. Survival tables were evaluated via Fisher’s exact test. A *p* value of less than 0.05 was regarded as indicating a statistically significant difference.

## 3. Results

### 3.1. Antibody Response Induced by Intramuscular Immunization

Mice immunized intramuscularly with 5 μg and 30 μg of HA-CCL19 DNA exhibited significantly higher antibody levels than those immunized with the same dose of HA DNA (Table 1). The IgG titer elicited by the HA-CCL19 DNA vaccine was approximately 16-fold greater than that provoked by the HA DNA vaccine at the same dose of 5 μg/mouse. Nevertheless, at a dose of 30 μg/mouse, the IgG titer engendered by the HA-CCL19 DNA vaccine was about 6.36-fold higher than that produced by the HA DNA vaccine. Two weeks post booster immunization, the IgG titer stimulated by the HA-CCL19 DNA vaccine was roughly 40-fold higher than that induced by the HA DNA vaccine at a dose of 5 μg/mouse. However, in the 30 μg/mouse immunization dose cohort, the level (titer) of IgG antibodies instigated by the HA-CCL19 DNA vaccine was approximately 32-fold greater than that incited by the HA DNA vaccine (Table 1).

Both the HA DNA vaccine and the HA-CCL19 DNA vaccine potently elicited the generation of hemagglutination inhibition (HI) antibodies subsequent to immunization. The HI antibody titers induced by the 5 μg and 30 μg HA-CCL19 DNA vaccines were 2^4.66^ and 2^6.33^, respectively, which were significantly higher than those of mice immunized with the same dose of the HA-CCL19 DNA vaccine. These findings demonstrated a statistically significant augmentation in the HI antibody levels stimulated by the HA-CCL19 DNA vaccine at the same dosages (*p* < 0.05) (Table 1). Moreover, it was observed that the HI antibody titers rose proportionally with the increasing immunization doses of both DNA vaccines, indicating a pronounced dose-dependent relationship between the HI antibody levels and the DNA vaccines.

Furthermore, we probed into the trend of IgG1 and IgG2a titers in the serum of mice subsequent to booster immunization. The results suggested that the titers of IgG1 and IgG2a antibodies escalated concomitant with the rising dose of the DNA vaccine. The ratio of IgG2a to IgG1 was greater than 1, which evinced that the integration of CCL19 did not adjust the immune predisposition of the DNA vaccine (Table 2).

### 3.2. Cellular Immune Responses Generated by Intramuscular Injection

Mice were immunized twice with the HA DNA vaccine and HA-CCL19 DNA vaccine by intramuscular injection with a two-week interval. Two weeks after the second immunization, spleen lymphocytes of immunized and control mice were isolated and stimulated with homologous influenza virus protein. The expression of IFN-γ and IL-2 in the culture medium supernatant were detected. At a dose of 5 μg/mouse, the level of IL-2 and IFN-γ increased about 1.3-fold and 1.69-fold in the HA-CCL19 DNA vaccine group, respectively (*p* < 0.05). At a dose of 30 μg/mouse, the levels of IFN-γ and IL-2 increased about 1.89-fold and 1.86-fold after the HA-CCL19 DNA vaccine, respectively (*p* < 0.05). Our results show that the expression of IL-2 and IFN-γ was significantly higher than that of the control group (Figure 2).

### 3.3. Protection Against Lethal Virus Challenge Provided by Intramuscular Injection

Mice were challenged a lethal dose of 5 LD_50_ of homologous influenza virus on the 14th day subsequent to booster immunization. The viral titers within the pulmonary tissues of the mice were determined on the third day post challenge. The survival rate and weight loss of the remaining mice were monitored over a 21-day period following the challenge.

The maximum weight loss rate of mice immunized with 5 μg/mouse and 30 μg/mouse of HA-ccl19 DNA vaccine was 11.2% and 10.2%, respectively. Meanwhile, the maximum weight loss rate of mice immunized with the HA DNA vaccine was significantly higher than that of mice immunized with the same doses of the HA-ccl19 DNA vaccine (Figure 3A). A statistically significant disparity in weight loss after the challenge was discerned between the two DNA vaccines (*p* < 0.05), suggesting that the CCL19 molecular adjuvant prominently alleviated the clinical manifestations in challenged mice and augmented the protective efficacy of the DNA vaccine. Notably, only 25% survival was recorded for mice immunized with a dose of 5 μg/mouse HA vaccine post challenge; conversely, survival attained an imposing 87.5% for those receiving an identical dosage of the HA-CCL19 DNA vaccine after the challenge. Remarkably, all mice vaccinated a dose of 30 μg per mouse endured a lethal dose challenge from the influenza virus with 100% survival (Figure 3B). These results suggested that CCL19 as a molecular adjuvant could markedly enhance the survival protection conferred by DNA vaccination in immunized mice (*p* < 0.05).

The viral titers in the lungs of mice immunized with a 5 μg/mouse HA DNA vaccine reached 10^4.00^ TCID_50_, whereas the viral titers in the lungs of mice receiving an equivalent dose of the HA-CCL19 DNA vaccine attained 10^2.67^ TCID_50_. When the immunization dose of the DNA vaccine was raised to 30 ug per mouse, the virus titer in the mice’s lungs was further reduced, suggesting that there was a dependence between the decrease of the virus in the mice’s lungs and the dose of the DNA vaccine (Figure 3C). The results from the post-challenge detection of viral titers indicated that the incorporation of CCL19 molecular adjuvant into the DNA vaccine significantly diminished viral loads in the lungs of mice receiving identical doses (*p* < 0.05).

### 3.4. Antibody Response Subsequent to Intranasal Mucosal Immunization

In light of the results derived from the intramuscular immunization approach adopted in this study, the HA-CCL19 DNA vaccine was singled out and administered at a dosage of 30 μg/mouse for the purpose of exploring the efficacy of the DNA vaccine through intranasal inoculation. No specific IgG was detected in the serum of mice that received the naked HA-CCL19 DNA vaccine. In contrast, mice immunized with the HA-CCL19-PEI composite exhibited an elevated IgG level (titer) following the initial immunization. Two weeks subsequent to booster immunization, the serum IgG titer underwent a further augmentation. The titers of IgG antibody elicited by the HA-CCL19-PEI-CS composite were significantly higher than those elicited by the HA-CCL19-PEI composite at both the primary and booster immunization stages (Table 3). Additionally, two weeks post booster vaccination, both IgG1 and IgG2a titers were measured in the serum, allowing for their ratio calculation. The results indicate that both IgG1 and IgG2a titers escalated with increasing doses of the DNA vaccine, mirroring the trend observed in total IgG levels. Notably, the ratio of specific IgG2a/IgG1 in sera from HA-CCL19-PEI-immunized mice was greater than 1 (Table 4).

Furthermore, the titers of specific sIgA in nasal lavage fluid (NLF) were determined and quantified. Upon primary immunization, no specific sIgAs were identified in the NLF of mice immunized with the naked HA-CCL19 DNA vaccine. Concurrently, in the nasal lavage fluid of mice immunized with the HA-CCL19-PEI composite and the HA-CCL19-PEI-CS (chitosan) composite, no specific sIgAs were detectable, either. Post booster immunization, a marked augmentation of sIgA was observed in both groups (*p* < 0.05), when compared with the group subjected to the naked HA-CCL19 DNA immunization. The sIgA titer elicited by the HA-CCL19-PEI-CS was approximately 5-fold higher than that induced by the HA-CCL19-PEI (*p* < 0.05). sIgA remained undetectable in the nasal lavage fluid of mice immunized with naked HA-CCL19, even after booster immunization (Table 5).

Mice were immunized intranasally, and HI antibody titers in the serum were determined two weeks subsequent to booster immunization. The naked HA-CCL19 DNA vaccine failed to evoke detectable HI antibodies; conversely, the PEI-mediated HA-CCL19 DNA vaccine successfully elicited HI antibodies with a titer of approximately 2^4.00^ post booster administration. Markedly, the HI antibody levels induced by the HA-CCL19 DNA vaccine delivered by both PEI and chitosan (CS) demonstrated a significant augmentation (*p* < 0.05), attaining a titer of approximately 2^5.33^ (Table 5).

### 3.5. Cellular Immune Responses Induced by Intranasal Inoculation

Cellular immune responses evoked by intranasal mucosal immunization were assayed two weeks subsequent to booster immunization. No conspicuous augmentation in cytokine expression was discerned in splenic lymphocytes from mice immunized with naked HA-CCL19. In contradistinction, there was a pronounced upsurge in the expression levels of IL-2 and IFN-γ within the supernatant of spleen lymphocytes derived from mice vaccinated with both the HA-CCL19-PEI complex and the HA-CCL19-PEI-CS composite. Significantly, when comparing HA-CCL19-PEI to its CS composite counterpart, IL-2 and IFN-γ concentrations escalated 2.56-fold and 1.88-fold, respectively (*p* < 0.05). Our findings suggest that intranasal administration of a DNA vaccine via PEI effectively activates systemic cellular immunity in mice; moreover, the incorporation of chitosan composite as an adjuvant significantly potentiates the immune response associated with HA-CCL19-PEI (*p* < 0.05) (Figure 4).

### 3.6. Protection Against Lethal Virus Challenge Provided by Intranasal Inoculation

The protective potency of the HA-CCL19/PEI/CS composite was assessed through a lethal challenge with the H7N9 virus. Mice were exposed to a lethal dose equivalent to 5 LD_50_ of the homologous influenza virus on the 14th day subsequent to booster immunization. In contrast to the naked HA-CCL19 DNA vaccine, the HA-CCL19-PEI composite conspicuously attenuated the maximal weight loss rate in immunized mice. Moreover, the HA-CCL19-PEI-CS composite bestowed enhanced protective benefits compared to HA-CCL19-PEI, resulting in a further diminution of the maximal post-challenge weight loss rate (*p* < 0.05) (Figure 5A).

Mice subjected to vaccinations with diverse vaccine formulations manifested pronounced variances in survival rates subsequent to a lethal H7N9 virus challenge. The survival proportion of mice inoculated with the HA-CCL19-PEI composite vaccine was a meager 12.5%, while that of mice administered with the HA-CCL19 naked DNA vaccine was nil. Remarkably, mice immunized with the HA-CCL19-PEI-CS complex achieved a complete 100% survival rate (Figure 5B).

Analysis of lung virus titers disclosed that mice vaccinated with the naked HA-CCL19 DNA vaccine had lung virus titers attaining 10^6.33^ TCID_50,_ whereas those administered with the HA-CCL19-PEI composite exhibited a significant decrement to 10^4.00^ TCID_50_ (*p* < 0.05). The lung virus titer for mice treated with the HA-CCL19-PEI-CS composite was determined at 10^3.00^ TCID_50_ (*p* < 0.05) (Figure 5C).

## 4. Discussion

Vaccination is currently recommended by the World Health Organization as the most efficacious strategy for the containment of infectious diseases such as influenza [3,28,29]. In recent years, immunomodulatory molecular adjuvants have drawn escalating attention in the realm of vaccine research and development. Molecular adjuvants can directly interface with the immune system and stimulate the proliferation and differentiation of pertinent immune cells, thereby evoking a swifter and more extensive immune response. Among the multiplicity of molecular adjuvants that have been disclosed, CCL19, also known as MIP-3 beta, functions as an immune homeostatic chemokine [30]. As a ligand of CCR7, CCL19 effectively instigates the β-inhibin-mediated phosphorylation and internalization of CCR7, exerting a cardinal role in activating lymphocyte immune functionalities and specifically augmenting the body’s cellular immunity [31,32]. The interaction between CCL19 and CCR7 also contributes to the liberation of antiviral-related cytokines (e.g., IFN-γ) from immune cells, facilitating T cell proliferation and antigen uptake by dendritic cells (DCs) [30,33]. The immune system has the capacity to evoke vigorous immune reactions even in scenarios of scant antigenic expression, thereby facilitating the body’s defense against pathogen invasion.

In this study, we designated CCL19 as the molecular adjuvant for the influenza virus hemagglutinin (HA) DNA vaccine. After the intramuscular immunization of mice, it was discerned that CCL19 could efficaciously potentiate the immune response elicited by the HA DNA vaccine at the same dosage. The levels of IgG induced by HA-CCL19 in intramuscularly immunized mice were conspicuously higher than those generated by the HA DNA vaccine alone (Table 1). HI antibodies are specific immunoglobulins produced in response to the stimulation by the HA protein of the influenza virus and serve as a crucial indicator for evaluating the immunogenicity of influenza vaccines containing HA [34]. HI antibody titers reflect, to a certain extent, the neutralizing capacity of antibodies induced by influenza vaccination against viral infection. In the present study, HI titers resulting from the administration of the HA-CCL19 DNA vaccine were significantly elevated compared to those from standard HA DNA immunization in mice receiving an identical dose (*p* < 0.05) (Table 1 and Table 2). The results of HI antibody titer assays and ELISA antibody titer measurements indicated that the incorporation of CCL19 as a molecular adjuvant markedly enhanced the humoral immunity stimulated by DNA vaccination and facilitated the secretion of antigen-specific antibodies by B lymphocytes. Additionally, the ratio of IgG2a to IgG1 was consistently greater than 1, which means the enhancement does not change the immune bias inherent in the DNA vaccine. Nevertheless, the specific mechanism demands further exploration and will be the focus of our subsequent research endeavors.

To explore the influence of the CCL19 molecular adjuvant on potentiating cellular immunity in DNA vaccines, spleen lymphocytes were isolated and cultivated from mice subsequent to immunization with two disparate DNA vaccine formulations. The concentrations of antigen-specific IFN-γ and IL-2 in the culture supernatants were assayed. Besides its direct inhibitory effect on viral replication, IFN-γ potentiates NK cell activity and regulates B cell functionality [35]. IL-2 acts as a crucial cytokine that plays a cardinal role in immune cell differentiation; it activates T cells, facilitates their proliferation, and effectively enhances the cytolytic activity of both NK cells and lymphokine-activated killer cells [36]. Our results demonstrated that in mice immunized with an equivalent dose of DNA vaccine, the HA-CCL19 DNA vaccine significantly augmented the expression levels of IFN-γ and IL-2 in the supernatants of spleen lymphocyte cultures compared to the HA DNA vaccine (*p* < 0.05). This finding manifested in the molecular adjuvant CCL19 prominently enhancing cellular immune responses by stimulating the production of IFN-γ and IL-2 and inducing Th1 responses (Figure 2).

In the challenge protection study subsequent to intramuscular immunization, mice inoculated with the HA-CCL19 DNA vaccine demonstrated a reduced magnitude of weight loss and a more expeditious weight recovery compared to those immunized with an equivalent dose of the HA DNA vaccine. This implies that at the same immunization dosage, the molecular adjuvant CCL19 effectively mitigates the rate of body weight loss and alleviates the injury inflicted by the influenza virus in experimental mice (Figure 3A). Notably, in mice administered a high dose (30 μg/mouse), a 100% survival rate was achieved for both vaccine groups when exposed to a lethal dose of the influenza virus. Moreover, immunization with the HA-CCL19 DNA vaccine at a dosage of 5 μg/mouse yielded a significantly lower mortality rate than that observed with the HA DNA vaccine at the same dosage (Figure 3B). The assessment of viral titers in the lungs indicated that the viral titers in the lungs of mice immunized with the HA-CCL19 DNA vaccine were significantly diminished under comparable immunization conditions (Figure 3C). The results of the study on immunization via intramuscular injection suggest that the CCL19 molecular adjuvant can conspicuously enhance the protective efficacy of the HA DNA vaccine and concurrently reduce the requisite amount of the DNA vaccine (Figure 3).

Beyond the conventional modality of intramuscular immunization, mucosal immunity constitutes the primary line of immune defense against respiratory pathogens and antigens, attributed to the copious presence of mucosa-associated lymphoid tissues and cells within the nasal mucosa [21,37]. With the progressive advancement of mucosal immunity research, nasal mucosal immunization is increasingly regarded as a promising vaccination modality. Influenza represents a respiratory infectious disorder caused by the influenza virus, posing a substantial threat to public health. The nasopharynx serves as the initial entry point for the influenza virus into the human body. Intranasal vaccination can evoke antigen-specific sIgA as the vaccine activates the mucosal immune system through nasal mucosa-associated lymphoid tissue (NALT) [38]. NALT functions as both a site for humoral and cellular immunity induction in the upper respiratory tract and a locale for the local, long-term production of specific antibodies. In clinical practice, achieving direct delivery of DNA through the nasal cavity to elicit an immune response is nearly infeasible due to both the limited residence time of vaccines in this area and suboptimal antigen absorption resulting from mechanical barriers present in the mucous membranes [39]. In 2011, Torrieri-Dramard, L. et al. demonstrated that DNA delivered via PEI exhibited efficiency in nasal immune gene delivery that was 1000 times greater than that of naked DNA alone, successfully inducing elevated levels of hemagglutinin-specific sIgA antibodies [40]. Our findings also indicated that a PEI composite significantly enhanced the immune response following the nasal administration of the DNA vaccine (Table 3).

As a naturally occurring cationic polysaccharide, chitosan has manifested pronounced mucosal adjuvant efficacy in previous studies [41]. Simultaneously, chitosan can interact with anionic mucin on the cell surface via its excessive cationic charges and be subsequently internalized by cells [42]. In this research, PEI was directly complexed with negatively charged DNA to form a multi-polymeric DNA-PEI complex, which was then coalesced with cationic chitosan through complex coacervation to generate a DNA-PEI-CS composite. The results of our study also indicated that the chitosan adjuvant could efficaciously enhance the immunogenicity of the vaccine.

The results of this study showed that the serum antibody levels induced by HA-CCL19 DNA vaccine after intramuscular immunization and nasal mucosal immunization were consistent. This evinces that the composite formed by the HA-CCL19 DNA vaccine when coated with PEI and chitosan could proficiently elicit the generation of IgG via intranasal immunization. Significantly, after encapsulation with CS, the immune efficacy was prominently enhanced (*p* < 0.05) (Table 3 and Table 4). Moreover, the alterations in IgG levels were analogous to those in the total IgG levels. Intriguingly, intranasal immunization did not change the immune bias of the HA DNA vaccine.

Mucosal immunity mediated by sIgA constitutes a pivotal factor in safeguarding against the influenza virus [21,37]. sIgA exhibits exceptional efficacy in impeding the invasion of the influenza virus at the mucosal portals [38]. In the present study, the local sIgA content within the lungs of mice underwent a remarkable escalation, indicating that the PEI composite potentiated the nasal mucosal immune response to the DNA vaccine. The sIgA concentration of HA-CCL19/PEI-CS composite exceeded that of HA-CCL19/PEI composite. Analogous outcomes were obtained regarding the HI antibody titers stimulated by the DNA vaccine administered through intranasal immunization. The HI antibody titers elicited by the composite formed by HA-CCL19 and PEI-CS were conspicuously higher than those of the composite formed by HA-CCL19 and PEI (Table 5).

DNA vaccines delivered intranasally also elicited systemic cellular immune responses [40]. Two weeks subsequent to booster immunization, splenic lymphocytes of immunized mice were isolated and subsequently subjected to antigenic stimulation. The cytokine levels in the supernatants of the cultured lymphocytes were assayed. The results demonstrated a marked augmentation in the concentrations of cytokines IFN-γ and IL-2, and this trend was in line with the results obtained from intramuscular immunization (Figure 4).

The results of the challenge protection experiment manifested that the mice immunized with HA-CCL19-PEI-CS composites underwent a certain extent of body weight decrement post challenge but displayed rapid recuperation (Figure 5A). Remarkably, the ultimate survival rate of these mice attained 100% throughout the entire observation period (Figure 5A). On the third day subsequent to the challenge, mice immunized with the naked HA-CCL19 DNA vaccine or the HA-CCL19-PEI composite manifested significantly higher viral titers in their lungs compared to those inoculated with the HA-CCL19-PEI-CS composite (Figure 5C). It is of paramount importance to note that although specific IgG and HI antibodies were detectable in mice immunized with the HA-CCL19-PEI composite, a significant reduction in the viral load in the lungs was also observed after the challenge when juxtaposed with those receiving only the HA-CCL19 DNA vaccine; however, during the entire observation period post challenge, their survival rate was merely 12.5% (Figure 5B). This accentuates the role of chitosan as an adjuvant in augmenting the immune effect of the vaccine; nevertheless, further exploration into the underlying mechanisms accountable for this result is imperative. Additionally, it should be noted that in the investigation of the HA DNA vaccine without the fusion expression of the molecular adjuvant CCL19, although specific IgG and HI antibodies along with specific activated cellular immunity were detectable in mice administered PEI and chitosan as adjuvants via the nasal mucosa, there was no improvement in either the rate of weight loss or the survival of the challenged mice when compared to both the exposed HA DNA-vaccinated groups and the PBS control groups.

Polyethyleneimine (PEI), a water-soluble cationic polymer, has garnered extensive utilization in the domains of gene transfection and drug delivery. In recent times, PEI has witnessed escalating employment as a non-viral vector for gene conveyance. The highly cationic nature of PEI enables electrostatic interactions with negatively charged cell membranes, thereby potentiating the cellular uptake of exogenous genes [43]. Chitosan (CS), a natural polysaccharide derived from chitin, has a broad spectrum of applications in drug, vaccine, and antigen delivery. Chitosan and its derivative composites exhibit substantial potential as adjuvants or delivery systems, capable of evoking potent humoral, cellular, and mucosal immune responses under diverse antigens and inoculation routes [42].

Currently, PEI is recognized as one of the most promising cationic carriers for achieving high transfection efficiency across various mammalian cell types. However, despite its numerous advantages as a DNA vaccine delivery system, the number of clinical applications to date has been limited due to its cytotoxicity [44]. The chitosan-based complexes developed to meet diverse requirements exhibit variations in molecular weight, morphology, particle size distribution, encapsulation efficiency, and drug loading capacity. Even when employing the same drug formulation, differences in manufacturing processes or operational techniques can result in discrepancies regarding drug encapsulation efficiency, loading capacity, and release rates [45,46]. Consequently, the mass production of many such formulations remains impractical at present.

In this investigation, a cationic PEI vector was utilized for the delivery of DNA, and a chitosan (CS) mucosal adjuvant was coated onto the PEI to explore the delivery efficiency and immune potentiation effect of the DNA vaccine. A drawback of this experiment pertains to the optimization issues in the fabrication of the DNA-PEI-CS composite. Disparities in the ratios among PEI, CS, and DNA can give rise to distinct diameters in the ultimately synthesized composite and variances in the in vivo transfection efficacy. The principal aim of this study was to evaluate the feasibility of intranasal immunization with a DNA vaccine against influenza virus. Hence, the method with the least synthesis volume was adopted to minimize the volume of the composite that was ultimately immunized in mice. Our results offer preliminary evidence that immunization with a DNA/PEI/CS composite via the nasal mucosal route can confer protection against lethal doses of influenza virus challenges. However, the advancement of effective delivery systems continues to be the foremost challenge in the field of DNA vaccine research. 

In future endeavors, we will optimize the ratios of DNA, PEI, and CS to determine the optimal combination of the three in the intranasal immunization route of the influenza DNA vaccine. We will strive to further refine the methodology for delivering DNA vaccines aimed at eliciting robust mucosal immunity, as well as explore alternative molecular adjuvants designed to enhance the immune response while judiciously minimizing the quantity of DNA utilized. Additionally, we are committed to pioneering innovative formulations of DNA vaccines that hold promise for improved efficacy and safety. Understanding the underlying mechanisms will also be a central focus of our future endeavors.

DNA vaccines boast a multitude of merits, such as expeditious preparation, prompt antigen updating to align with emerging variants, the capacity to manufacture vaccines within the shortest timeframes, facile storage and transportation, and the potential to evoke both humoral and cellular immunity [5]. Nevertheless, with the sole exception of one sanctioned DNA product for COVID-19 in India, no other DNA vaccine offerings have been employed. Despite the distinct advantages that DNA vaccines hold over conventional vaccines, numerous issues still await resolution, particularly the enhancement of the immunogenicity of DNA vaccines [20]. The results of this study demonstrated that molecular adjuvants can augment the immunogenicity of DNA vaccines and validate the efficacy and feasibility of intranasal influenza DNA vaccination. This provides a novel concept for the development of influenza vaccines and furnishes a new armamentarium to contend with potential influenza pandemics.

## 5. Conclusions

CCL19, functioning as a molecular adjuvant, can conspicuously augment the immunogenicity of DNA vaccines. The composite engendered by the association of DNA vaccines with cationic vectors such as PEI and CS are competent at eliciting both local mucosal immunity and systemic immunity subsequent to nasal administration. This study has provided valuable insights that can facilitate advancements in the development of nasal delivery systems for DNA vaccines targeting influenza or other diseases.

## Figures and Tables

**Figure 1 vaccines-13-00010-f001:**
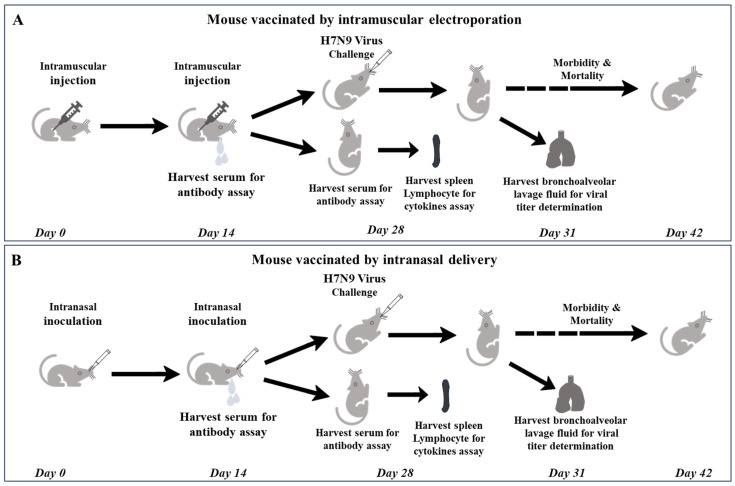
Schematic diagram of schedule of DNA vaccine immunization and following bioassays and challenges with virus. (**A**) mouse vaccinated by intramuscular electroporation; (**B**) mouse vaccinated by intranasal delivery.

**Figure 2 vaccines-13-00010-f002:**
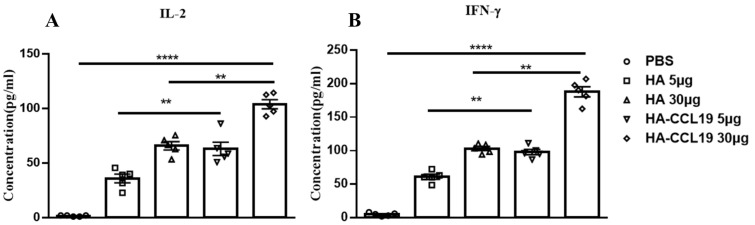
T cell immune response elicited by either the HA DNA vaccine or the HA-CCL19 DNA vaccine through the intramuscular administration approach. Five mice in each group were intramuscularly immunized with disparate doses of either the HA DNA vaccine or the HA-CCL19 DNA vaccine, as delineated. The interval between primary immunization and booster immunization was two weeks. Two weeks subsequent to booster immunization, spleen lymphocytes of immunized and control mice were isolated and cultivated. The expression levels of IL-2 (**A**) or IFN-γ (**B**) in the supernatant of the culture medium were determined by ELISA. A one-way analysis of variance (ANOVA) was utilized for the statistical significance assessment between the vaccinated cohorts and the control group. **** *p* < 0.0001 represents a significant difference between the vaccinated group and the control group. ** *p* < 0.01 represents a significant difference between the HA group and the HA-CCL19 group.

**Figure 3 vaccines-13-00010-f003:**
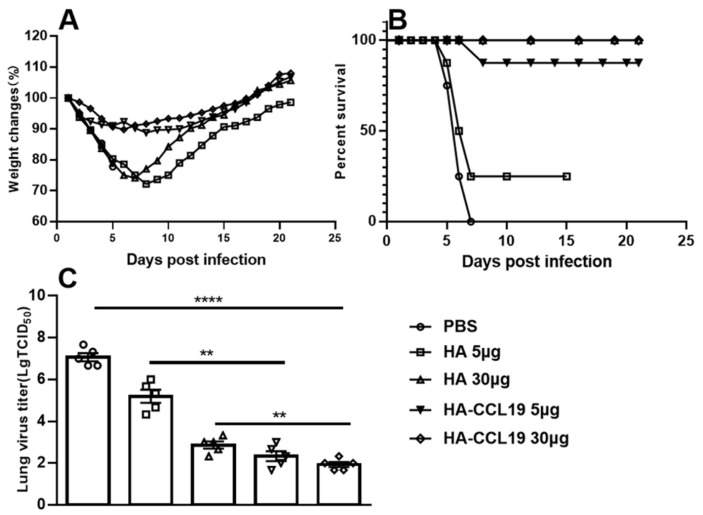
The protective efficacy of diverse DNA vaccines against the challenge of lethal influenza virus subsequent to the intramuscular administration approach. As depicted in the figure, the HA DNA or HA-CCL19 DNA vaccine was administered to mice at dosages of 5 µg or 30 µg, respectively, while the control mice were immunized with an empty plasmid in the same manner, with 15 mice in each group. The interval between primary immunization and booster immunization was two weeks. Two weeks following the booster immunization, the mice were challenged with lethal H7N9 virus at a dose of 5 LD_50_. On the third day after the challenge, five mice from each group were randomly selected for the collection of lung tissue and the determination of the viral titer in the lung tissue; the rest of the mice were observed daily for the morbidity record. When the mice underwent a body weight reduction surpassing 25%, they were subjected to euthanasia. (**A**): body weight loss; (**B**): mortality rates; (**C**): viral titers in the lung tissue. A one-way analysis of variance (ANOVA) was utilized for the statistical significance assessment between the vaccinated cohorts and the control group. **** *p* < 0.0001 represents a significant difference between the vaccinated group and the control group. ** *p* < 0.01 represents a significant difference between the HA group and the HA-CCL19 group.

**Figure 4 vaccines-13-00010-f004:**
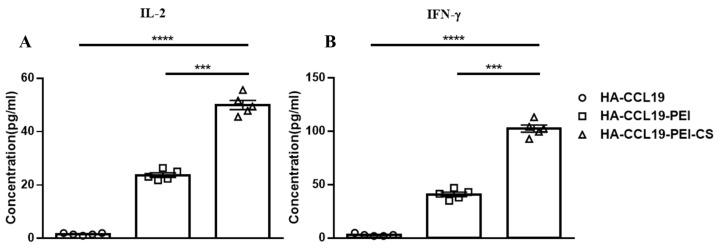
T cell immune response elicited by the HA-CCL19 DNA vaccine or the HA-CCL19/PEI or HA-CCL19/PEI/CS composite through the intranasal administration approach. Five mice in each group were intranasally immunized with the HA-CCL19 DNA vaccine, HA-CCL19/PEI or HA-CCL19/PEI/CS composite, as delineated. The application dose of HA-CCL19 in different formulas was 30 µg per mouse. The interval between primary immunization and booster immunization was two weeks. Two weeks subsequent to booster immunization, spleen lymphocytes of immunized and control mice were isolated and cultivated. The concentration of IL-2 (**A**) or IFN-γ (**B**) in the supernatant of the culture medium were determined by ELISA. A one-way analysis of variance (ANOVA) was utilized for the statistical significance assessment between the vaccinated cohorts and the control group. **** *p* < 0.0001 represents a significant difference between the HA-CCL19/PEI or HA-CCL19/PEI/CS composite-vaccinated group and the naked HA-CCL19 DNA vaccine group. *** *p* < 0.001 represents a significant difference between the HA-CCL19/PEI group and the HA-CCL19/PEI/CS group.

**Figure 5 vaccines-13-00010-f005:**
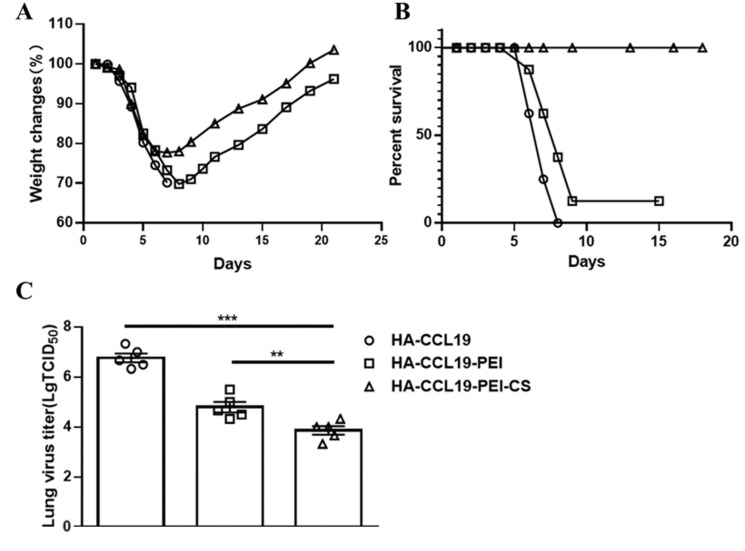
The protective efficacy of the HA-CCL19 DNA vaccine and the HA-CCL19/PEI or HA-CCL19/PEI/CS composite against the challenge of lethal influenza virus subsequent to the intranasal administration approach. Fifteen mice in each group were vaccinated with the HA-CCL19 DNA vaccine or the HA-CCL19/PEI or HA-CCL19/PEI/CS composite, respectively. The application dose of HA-CCL19 in different formula was 30 µg per mouse. The interval between primary immunization and booster immunization was two weeks. Two weeks following the booster immunization, the mice were challenged with a lethal H7N9 virus at a dose of 5 LD_50_. On the third day after challenge, five mice from each group were randomly selected for the collection of lung tissue and the determination of the viral titer in the lung tissue; the rest of the mice were observed daily for the morbidity record. When the mice underwent a body weight reduction surpassing 25%, they were subjected to euthanasia. (**A**): body weight loss; (**B**): mortality rates; (**C**): viral titers in the lung tissue. A one-way analysis of variance (ANOVA) was utilized for the statistical significance assessment between the vaccinated cohorts and the control group. *** *p* < 0.001 represents a significant difference between the HA-CCL19/PEI or HA-CCL19/PEI/CS composite-vaccinated group and the naked HA-CCL19 DNA vaccine group. ** *p* < 0.01 represents a significant difference between the HA-CCL19/PEI group and the HA-CCL19/PEI/CS group.

**Table 1 vaccines-13-00010-t001:** Antibody titers in mice immunized with HA and HA-CCL19 DNA vaccines.

Group	Serum IgG Titer (2^n^) After First Immunization ^a^	Serum IgG Titer (2^n^) After Second Immunization ^a^	HI Titer (2^n^) ^a^
PBS	―	―	ND
HA 5 μg	6.33 ± 0.58 ^b,c^	12.67 ± 0.58 ^b,c^	3.33 ± 1.52 ^b,c^
HA 30 μg	11.33 ± 0.58 ^b,c,d^	18.33 ± 0.58 ^b,c,d^	4.66 ± 1.52 ^b,c^
HA-CCL19 5 μg	10.33 ± 0.58 ^b,c,d^	18.00 ± 1.00 ^b,c,d^	4.66 ± 0.58 ^b,c^
HA-CCL19 30 μg	14.00 ± 1.00 ^b,c,d,e^	23.33 ± 1.53 ^b,c,d,e^	6.33 ± 1.15 ^b,c^

Five mice in each group were intramuscularly immunized with disparate doses of either the HA DNA vaccine or the HA-CCL19 DNA vaccine, as delineated. Two weeks after the first and second immunization, blood samples were taken from tail vein of mice for measurement of specific IgG titer in sera by ELISA, and blood samples were collected two weeks after the second immunization to measure HI titer. ^a^ Results are expressed as the mean ± SD of each group. ^b^ Significant difference compared with the control group, *p* < 0.05. ^c^ Significant difference compared with the control group, *p* < 0.05. ^d^ Significant difference compared with group HA 5 μg, *p* < 0.05. ^e^ Significant difference compared with group HA 30 μg, *p* < 0.05. ND: not detected.

**Table 2 vaccines-13-00010-t002:** Titers of IgG subclass antibodies in mice after immunization twice with HA and HA-CCL19 DNA vaccines.

Groups	IgG1 (2^n^) ^a^	IgG2a (2^n^) ^a^	IgG2a/IgG1
PBS	―	―	―
HA 5 μg	9.67 ± 0.58 ^b^	10.67 ± 2.08 ^b^	2 ± 0.35
HA 30 μg	15.33 ± 0.58 ^b^	16.33 ± 0.58 ^b^	2 ± 0.16
HA-CCL19 5 μg	12.33 ± 2.08 ^b^	15.67 ± 1.53 ^b^	10 ± 0.43
HA-CCL19 30 μg	17.33 ± 1.15 ^b^	20.00 ± 1.00 ^b^	6 ± 0.27

Five mice in each group were intramuscularly immunized with disparate doses of either the HA DNA vaccine or the HA-CCL19 DNA vaccine, as delineated. Two weeks after the first and second immunization, blood samples were taken from tail vein of mice for the measurement of specific IgG1 and IgG2a titer in sera by ELISA. Note: ^a^ Results are expressed as the mean ± SD of each group. ^b^ Significant difference compared with the PBS (control) group, *p* < 0.05.

**Table 3 vaccines-13-00010-t003:** Antibody titers in mice immunized with HA-CCL19/DNA/PEI/CS composite.

Groups	Serum IgG Titer (2^n^) After First Immunization ^a^	Serum IgG Titer (2^n^) After Second Immunization ^a^
PBS	―	―
HA-CCL19	ND	ND
HA-CCL19-PEI	8.66 ± 0.58 ^b^	10.66 ± 1.15 ^b^
HA-CCL19-PEI-CS	11.66 ± 0.58 ^b,c^	15.00 ± 1.00 ^b,c^

Note: Five mice in each group were intranasally immunized with the HA-CCL19 DNA vaccine or the HA-CCL19/PEI or HA-CCL19/PEI/CS composite, as delineated. The application dose of HA-CCL19 in different formula was 30 µg per mouse. Two weeks after the first and second immunization, blood samples were taken from the tail veins of the mice for the measurement of the specific IgG titer in sera by ELISA. ^a^ Results are expressed as the mean ± SD of each group. ^b^ Significant difference compared with group HA-CCL19, *p* < 0.05. ^c^ Significant difference compared with group HA-CCL19-PEI, *p* < 0.05. ND: not detected.

**Table 4 vaccines-13-00010-t004:** Titers of IgG subclass antibodies in mice after being immunized twice with HA-CCL19/PEI/CS composite.

Groups	IgG1 (2^n^) ^a^	IgG2a (2^n^) ^a^	IgG2a/IgG1
PBS	―	―	―
HA-CCL19	ND	ND	ND
HA-CCL19-PEI	8.33 ± 0.58 ^b^	9.33 ± 0.58 ^b^	2 ± 0.33
HA-CCL19-PEI-CS	11.67 ± 0.58 ^b,c^	14.00 ± 0.00 ^b,c^	5 ± 0.16

Note: Five mice in each group were intranasally immunized with the HA-CCL19 DNA vaccine or the HA-CCL19/PEI or HA-CCL19/PEI/CS composite, as delineated. The application dose of HA-CCL19 in different formulas was 30 µg per mouse. Two weeks after the first and second immunization, blood samples were taken from the tail veins of the mice for the measurement of the specific IgG titer in sera by ELISA. ^a^ Results are expressed as the mean ± SD of each group. ^b^ Significant difference compared with group HA-CCL19, *p* < 0.05. ^c^ Significant difference compared with group HA-CCL19-PEI, *p* < 0.05. ND: not detected.

**Table 5 vaccines-13-00010-t005:** Titer of sIgA after being immunized twice with PEI/CS composite.

Groups	Serum sIgA titer (2^n^) ^a^	HI titer (2^n^) ^a^
PBS	―	ND
HA-CCL19	ND	ND
HA-CCL19-PEI	5.66 ± 1.15 ^b,c^	4.00 ± 1.00 ^b^
HA-CCL19-PEI-CS	8.00 ± 1.00 ^b,c,d^	5.33 ± 0.58 ^b,c^

Note: Two weeks after being immunized twice, the lung was taken for the measurement of specific sIgA titer by ELISA, and blood samples were collected for the HI titer. ^a^ Results are expressed as the mean ± SD of each group. ^b^ Significant difference compared with group HA, *p* < 0.05. ^c^ Significant difference compared with group HA-PEI, *p* < 0.05. ^d^ Significant difference compared with group HA-CCL19-PEI, *p* < 0.05. ND: not detected.

## Data Availability

Data are contained within the article. The raw data supporting the conclusions of this article will be made available by the authors on request.

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
