# Peer review of "Intranasal Immunization with DNA Vaccine HA-CCL19/Polyethylenimine/Chitosan Composite Provides Immune Protection Against H7N9 Infection"

_vaccines, 2024, doi:10.3390/vaccines13010010_

Round 1
Reviewer 1 Report
Comments and Suggestions for Authors
The manuscript entitled "Intranasal Immunization with DNA Vaccine HA-CCL19/Polyethylenimine/chitosan composite provides immune protection against homologous H7N9 Infection" presents a significant contribution to the field of vaccine development by exploring the use of molecular adjuvant CCL19 and innovative delivery methods (PEI and chitosan) to enhance the efficacy of DNA vaccines against H7N9 avian influenza. This approach has clear implications for improving immune responses, particularly through mucosal routes.
However, addressing the minor gaps could enhance the impact and appeal of the study to a broader audience.
1. Abstract: While informative, the abstract could better emphasize the broader implications of the findings for public health and future research directions.
2. Introduction: The introduction could provide a deeper review of previous research to better highlight the novelty of the study.
3. Methodology: it clearly describes the experimental design, including the use of molecular adjuvants and both intramuscular and intranasal routes of administration. Some sections, such as the fabrication of the DNA-PEI-CS composite, lack sufficient detail for reproducibility. Statistical methods are described but could be expanded to include a rationale for the chosen analysis techniques.
4. Results: Comprehensive data presentation, including antibody titers, cytokine levels, and survival rates, demonstrates robust immune responses elicited by the vaccine. Graphs and tables are well-organized and provide clear visual evidence of the findings.
5. Discussion : It occasionally reiterates results rather than providing deeper insight or critical analysis. While limitations are mentioned, they are not explored in sufficient depth to inform future research strategies.
6. Conclusions: The conclusions could better address the limitations of the study, such as challenges in scaling the DNA-PEI-CS composite fabrication for clinical use.
Author Response
- Abstract: While informative, the abstract could better emphasize the broader implications of the findings for public health and future research directions.
Thank you very much for your comment. Owing to the character limitation in the abstract section, we are unable to incorporate extensive content. According to your suggestion, we have added this information in the abstract section.
“This study investigated the feasibility of utilizing nasal mucosa for DNA vaccine immunization, which holds significant implications for the advancement and application of DNA vaccines in public health.” See this change in Page 1, line 27-29.
- Introduction: The introduction could provide a deeper review of previous research to better highlight the novelty of the study.
Thank you for your comment. According to your suggestion. We have added these sentences in the introduction section (see this change in page 2, line 55-63).
“Chitosan (CS) and polyethyleneimine (PEI) exhibit bio-adhesive and immunomodulatory properties, making them widely utilized as adjuvants and delivery vectors in the field of mucosal immunity research. Both polyethyleneimine (PEI) and chitosan can bind to proteoglycans on the cell surface, facilitating their entry into cells via endocytosis. Previous studies have demonstrated that PEI-modified chitosan exhibits high transfection efficiency in vivo. Intranasal immunization offers several advantages over intramuscular immunization. This method demonstrates high immunization efficiency, strong patient compliance, and reduced vaccination costs. The investigation into the integration of the advantages offered by intranasal immunization with DNA vaccines carries substantial implications for public health.”
- Methodology: it clearly describes the experimental design, including the use of molecular adjuvants and both intramuscular and intranasal routes of administration. Some sections, such as the fabrication of the DNA-PEI-CS composite, lack sufficient detail for reproducibility. Statistical methods are described but could be expanded to include a rationale for the chosen analysis techniques.
Thank you very much for your comment. According to your suggestion. We have added these sentences in the materials and method section to describe the fabrication of the DNA-PEI-CS composite.
“PEI was dissolved in ultrapure water to achieve a final concentration of 1 mg/ml. An aqueous solution of 1% acetic acid was formulated firstly, and subsequently, a 0.1% chitosan solution (mass-to-volume ratio) was prepared with the 1% aqueous acetic acid solution. Firstly, the required PEI aqueous solution was prepared based on the total amount of DNA needed for immunizing the mice, adhering to a PEI/DNA N/P ratio of 10. (N/P refers to the ratio of moles of amine groups in cationic polymers to those of phosphate groups in DNA.) The N/P ratio is calculated by the formula: 7.53×b/c, where b represents the mass of PEI (μg) and c denotes the mass of plasmid (μg). This mixture was then added to the DNA aqueous solution, where N signifies the amino groups of PEI and P indicates the phosphate groups of DNA. The components were thoroughly mixed with DNA and allowed to stand at room temperature for 30 minutes to facilitate the synthesis of the DNA-PEI nanocomposite. Subsequently, this complex was preheated to 55°C in conjunction with a 0.1% CS solution. The corresponding composite were obtained by rapidly adding an equal volume of CS solution to the DNA-PEI complex. The mixture was vortexed for 30 seconds and left at room temperature for an additional 30 minutes.” This change can be found in page 3, line 112-128.
According to your suggestion, we present the rationale for the selected analysis techniques both here and in the manuscript: In this study, serum antibody titers and weight loss rates were subjected to one-way analysis of variance (ANOVA) in GraphPad Prism, when there is only one independent variable with more than two levels or groups. One-way ANOVA was used to assess whether there are any statistically significant differences among the means of the immunized mice groups and control group. The survival rates of different groups were appraised by the log-rank test. The log-rank test is a non-parametric method that does not assume a specific distribution for survival time. Instead, it compares the entire distribution of survival times rather than solely evaluating the survival rate at a particular time point. Survival tables were evaluated via Fisher's exact test, and a P value less than 0.05 was regarded as indicating a statistically significant difference. The Fisher exact probability method is fundamentally a statistical approach that enables the direct calculation of probabilities based on the principles of hypergeometric distribution. While this method does not fall within the purview of chi-square tests, it serves as a complementary testing technique to enhance the robustness of chi-square analyses. The corresponding changes can be found in page 4, line 151-153.
- Results: Comprehensive data presentation, including antibody titers, cytokine levels, and survival rates, demonstrates robust immune responses elicited by the vaccine. Graphs and tables are well-organized and provide clear visual evidence of the findings.
Thank you very much for your comment.
- Discussion: It occasionally reiterates results rather than providing deeper insight or critical analysis. While limitations are mentioned, they are not explored in sufficient depth to inform future research strategies.
Thank you for your comment. According to your suggestion, we have added the following information in the discussion section of the manuscript.
"Currently, PEI is recognized as one of the most promising cationic carriers for achieving high transfection efficiency across various mammalian cell types. However, despite its numerous advantages as a DNA vaccine delivery system, the number of clinical applications to date has been limited due to its cytotoxicity. The chitosan-based complexes developed to meet diverse requirements exhibit variations in molecular weight, morphology, particle size distribution, encapsulation efficiency, and drug loading capacity. Even when employing the same drug formulation, differences in manufacturing processes or operational techniques can result in discrepancies regarding drug encapsulation efficiency, loading capacity, and release rates. Consequently, mass production of many such formulations remains impractical at present." This change can be found in page 14, line 566-575.
"Our results offer preliminary evidence that immunization with a DNA/PEI/CS composite via the nasal mucosal route can confer protection against lethal doses of influenza virus challenge. However, the advancement of effective delivery systems continues to be the foremost challenge in the field of DNA vaccine research. However, the advancement of effective delivery systems continues to be the foremost challenge in the field of DNA vaccine research." This change can be found in page 14, line 585-590."We will strive to further refine the methodology for delivering DNA vaccines aimed at eliciting robust mucosal immunity, as well as explore alternative molecular adjuvants designed to enhance the immune response while judiciously minimizing the quantity of DNA utilized. Additionally, we are committed to pioneering innovative formulations of DNA vaccines that hold promise for improved efficacy and safety. Understanding the underlying mechanisms will also be a central focus of our future endeavors." This change can be found in page 14, line 593-598.
- Conclusions: The conclusions could better address the limitations of the study, such as challenges in scaling the DNA-PEI-CS composite fabrication for clinical use.
Thank you very much for your comment. Currently, polyethyleneimine (PEI) is recognized as one of the most promising cationic carriers for achieving high transfection efficiency across various mammalian cell types. However, despite its numerous advantages as a DNA vaccine delivery system, the number of clinical applications to date has been limited. The chitosan-based complexes developed to meet diverse requirements exhibit variations in molecular weight, morphology, particle size distribution, encapsulation efficiency, and drug loading capacity. Even when employing the same drug formulation, differences in manufacturing processes or operational techniques can result in discrepancies regarding drug encapsulation efficiency, loading capacity, and release rates. Consequently, mass production of many such formulations remains impractical at present.
In future research endeavors, we aim to reduce the DNA dosage while further optimizing the ratio of DNA/PEI/CS to enhance transfection efficiency in vivo while simultaneously improving immune protection. In addition to combining PEI-DNA with chitosan, we will also investigate the fabrication of various complexes using alternative materials to explore their immune-protective effects.
In conclusion, this study has provided valuable insights that can facilitate advancements in the development of nasal delivery systems for DNA vaccines targeting influenza or other diseases.
We have added related information in the discussion section of the manuscript, please see the changes listed in commment 5 and we also made somse changes in the conclusion section of the manuscript.
”This study has provided valuable insights that can facilitate advancements in the development of nasal delivery systems for DNA vaccines targeting influenza or other diseases.“ This change can be found in page 15, line 615-617.
Reviewer 2 Report
Comments and Suggestions for Authors
Review comments on vaccines-3331844
Manuscript ID:  vaccines-3331844
Type:  Article
Title:  Intranasal Immunization with DNA Vaccine HA-CCL19/Polyethylenimine/chitosan composite provides immune protection against homologous H7N9 Infection
Authors:  Yuqing Xiang, Hongbo Zhang *, Youcai An, Ze Chen *
Section: Cellular/Molecular Immunology
Special Issue: Novel Viral Vaccine and Molecular Immunology
Major comments:
(1) In the present study, authors investigated a murine model to explore the molecular adjuvant efficacy of CCL19 in the context of DNA vaccination. Authors used a mouse CCL19 gene as a molecular adjuvant, which was concatenated to the hemagglutinin (HA) gene of H7N9 virus in the pCACCG vector. Authors used further polyethylene imine (PEI) and chitosan (CS) as parts of adjuvants for the delivery of the H7N9-HA-CCL19 DNA vaccine through the nasal mucosal route.
(2) The authors found that the CCL19 molecular adjuvant exerted a substantial immunomodulatory enhancement effect on the H7N9-HA DNA vaccine via intramuscularly immunization. Then, the authors showed that the efficacy of the H7N9-HA-CCL19 DNA vaccine through intranasal mucosal inoculation was significantly enhanced by delivering in the form of H7N9-HA-CCL19 DNA-PEI-CS composite. These observations were very novel and significant and validated the efficacy and feasibility of intranasal influenza DNA vaccination. Accordingly, this manuscript can become acceptable if appropriate responses and answers were made against following major and minor comments.
(3) The usage of CCL19 gene would be the most important point in the present study and some introductory description was made in the introduction section. However, in the introduction section, there is no information why the authors used PEI and CS in the present study. It might be better to include some information why PEI and chitosan were used.
(4) It might be better for easier understanding of readers if appropriate figures depicting the schedule of DNA vaccine immunization and following bioassays and the challenges with virus were added in the Materials and Methods section.
Minor comments:
(1) (Page 1, line 21) “H79” should be changed to “H7N9”.
(2) (Page 3, line 127) What is H7N9 protein? Please specify.
(3) (Page 3, line142)“8-fold higher” should be “6.36-fold higher” based on my own calculation. Please confirm.
(4) It is not clear that the description in lines 149-157 (page 4) is a legend to Table 1. If so, an appropriate space is required just after the line 157.
(5) (Page 4, line 186) “Figure1” should be “Figure 1”.
(6) It might be better to move Table 2 (page 6) just after the Table 1 (page 4).
(7) (Page 7, line 271) The title of Table 3 should be changed as “Antibody titers in mice immunized with HA-CCL19/PEI/CS composite” for clarification.
(8) (Page 8, line 324) “DNA-PEI” should be changed to “HA-CCL19-PEI” for clarification.
(9) (Page 9, line 342) “DNA/PEI/CS composite” should be changed to “HA-CCL19/PEI/CS composite” for clarification.
(10)(Page 12, lines 489-494) Table 5 and its legend should appear at page 8. It is very unusual that experimental results mentioned in page 8 were allocated in Discussion section.

none.
Author Response
Major comments:
(1) In the present study, authors investigated a murine model to explore the molecular adjuvant efficacy of CCL19 in the context of DNA vaccination. Authors used a mouse CCL19 gene as a molecular adjuvant, which was concatenated to the hemagglutinin (HA) gene of H7N9 virus in the pCACCG vector. Authors used further polyethylene imine (PEI) and chitosan (CS) as parts of adjuvants for the delivery of the H7N9-HA-CCL19 DNA vaccine through the nasal mucosal route.
We are extremely grateful for your comments.
(2) The authors found that the CCL19 molecular adjuvant exerted a substantial immunomodulatory enhancement effect on the H7N9-HA DNA vaccine via intramuscularly immunization. Then, the authors showed that the efficacy of the H7N9-HA-CCL19 DNA vaccine through intranasal mucosal inoculation was significantly enhanced by delivering in the form of H7N9-HA-CCL19 DNA-PEI-CS composite. These observations were very novel and significant and validated the efficacy and feasibility of intranasal influenza DNA vaccination. Accordingly, this manuscript can become acceptable if appropriate responses and answers were made against following major and minor comments.
We are extremely grateful for your comments.
3. The usage of CCL19 gene would be the most important point in the present study and some introductory description was made in the introduction section. However, in the introduction section, there is no information why the authors used PEI and CS in the present study. It might be better to include some information why PEI and chitosan were used.
Thank you very much for your valuable comment. We have incorporated the following content into the introduction section of the manuscript.
"Chitosan (CS) and polyethyleneimine (PEI) exhibit bio-adhesive and immunomodulatory properties, making them widely utilized as adjuvants and delivery vectors in the field of mucosal immunity research. Both PEI and CS can bind to proteoglycans on the cell surface, facilitating their entry into cells via endocytosis. Previous studies have demonstrated that PEI-modified chitosan exhibits high transfection efficiency in vivo. Intranasal immunization presents several advantages over intramuscular immunization, including enhanced immunization efficiency and improved patient compliance. The investigation of integrating the benefits provided by intranasal immunization with DNA vaccines holds significant implications for public health." This change can be found in page 2, line 55-63.
(4) It might be better for easier understanding of readers if appropriate figures depicting the schedule of DNA vaccine immunization and following bioassays and the challenges with virus were added in the Materials and Methods section.
Thank you very much for your comment. In accordance with your suggestion, we have incorporated a diagram depicting the schedule of DNA vaccine immunization and following bioassays and the challenges with virus were added in the Materials and Methods section of the manuscript.This change can be found in page 3, line 107-110.
Figure 1. Schematic diagram of the schedule of DNA vaccine immunization and following bioassays and the challenges with virus. A: mouse vaccinated by intramuscular electroporation; B: mouse vaccinated by intranasal delivery.
Minor comments:
(1) (Page 1, line 21) “H79” should be changed to “H7N9”.
We thank you very much for pointing out this error. We have changed "H79" to "H7N9”in the manuscript. This change can be found in page 1, line 21.
(2) (Page 3, line 127) What is H7N9 protein? Please specify.
We thank you very much for pointing this out. H7N9 protein used in this study is the lysate of purified H7N9 virus. This change can be found in page 4, line 146.
(3) (Page 3, line142)“8-fold higher” should be “6.36-fold higher” based on my own calculation. Please confirm.
We thank you very much for pointing this out. It should be 6.36-fold higher. We have changed this point in the manuscript. This change can be found in page 4, line 162.
(4) It is not clear that the description in lines 149-157 (page 4) is a legend to Table 1. If so, an appropriate space is required just after the line 157.
We thank you very much for pointing this out. The description in lines 149-157 (page 4) is a legend to Table 1. This is supposed to be a typesetting error, and we will notify the editorial department of the magazine responsible for the typesetting to rectify this.
(5) (Page 4, line 186) “Figure1” should be “Figure 1”.
We thank you very much for pointing out this error. We have changed "Figure1" to "Figure 2" in the manuscript. This change can be found in page 5, line 215. Due to the inclusion of a new image (Figure 1, located in the Materials and Methods section), the numbering of the original images has been adjusted accordingly.
(6) It might be better to move Table 2 (page 6) just after the Table 1 (page 4).
We thank you very much for pointing this out. we will notify the editor to move Table 2 (page 6) just after the Table 1 (page 4).
(7) (Page 7, line 271) The title of Table 3 should be changed as “Antibody titers in mice immunized with HA-CCL19/PEI/CS composite” for clarification.
We thank you very much for pointing out this error. We have changed the title of Table 3 to "Antibody titers in mice immunized with HA-CCL19/PEI/CS composite". This change can be found in page 8, line 301.
(8) (Page 8, line 324) “DNA-PEI” should be changed to “HA-CCL19-PEI” for clarification.
We thank you very much for pointing this out. We have changed "DNA-PEI" to "HA-CCL19-PEI" in the manuscript. This change can be found in page 9, line 355.
(9) (Page 9, line 342) “DNA/PEI/CS composite” should be changed to “HA-CCL19/PEI/CS composite” for clarification.
We thank you very much for pointing this out. We have changed "DNA/PEI/CS composite" to "HA-CCL19/PEI/CS composite" in the manuscript. This change can be found in page 10, line 374.
(10) (Page 12, lines 489-494) Table 5 and its legend should appear at page 8. It is very unusual that experimental results mentioned in page 8 were allocated in Discussion section.
We thank you very much for pointing this out. We will remind the typesetting editor to correct this error.
Reviewer 3 Report
Comments and Suggestions for Authors
One of the keywords in this manuscript is HA-CCL19/Poly-ethylenimine/chitosan. The investigators applied creative methods to develop the antibody and cytokines-producing vaccine against AIV H7N9.
This reviewer is EXTREMELY concerned about the inconsistent data in this manuscript. In Figure 2, the HA-CCL 19 groups showed survival rates of 90% (5 micrograms) and 100% (30 micrograms). However, the same group in Figure 4 demonstrated a survival rate of 0% on day 8 or 9 after challenge infection.
Please consider removing "homologous" from the manuscript title.
Author Response
Comment 1:This reviewer is EXTREMELY concerned about the inconsistent data in this manuscript. In Figure 2, the HA-CCL 19 groups showed survival rates of 90% (5 micrograms) and 100% (30 micrograms). However, the same group in Figure 4 demonstrated a survival rate of 0% on day 8 or 9 after challenge infection.
Response 1: We sincerely appreciate you bringing this to our attention. In Figure 2, the HA-CCL19 groups exhibited survival rates of 90% (5 micrograms per mouse) and 100% (30 micrograms per mouse). In this experiment, the mice were immunized via intramuscular injection and electroporation. However, in the experiment depicted in Figure 4, the mice were immunized through an intranasal route. The survival rate of mice immunized with HA-CCL19 naked DNA vaccine by intranasal inoculation was 0, while the survival rate of mice immunized with HA-CCL19-PEI composite was 12.5%, and the survival rate of mice immunized with HA-CCL19-PEI-CS complex was 100% It is essential to highlight that the intranasal administration of naked HA-CCL19 DNA does not induce an immune response. Moreover, even the HA-CCL19/PEI composite fails to elicit an effective immune reaction when administered via intranasal inoculation. It is important to note that different immunization methods and composite can yield varying results.
Comment 2: Please consider removing "homologous" from the manuscript title.
We thank you very much for pointing this out. We will remove "homologous" from the manuscript title.
Round 2
Reviewer 1 Report
Comments and Suggestions for Authors
The authors have satisfactorily responded to my comments.
Reviewer 3 Report
Comments and Suggestions for Authors
Thanks for the revision.